# Circadian Variation in Human Milk Composition, a Systematic Review

**DOI:** 10.3390/nu12082328

**Published:** 2020-08-04

**Authors:** Merel F. Italianer, Eva F. G. Naninck, Jorine A. Roelants, Gijsbertus T. J. van der Horst, Irwin K. M. Reiss, Johannes B. van Goudoever, Koen F. M. Joosten, Inês Chaves, Marijn J. Vermeulen

**Affiliations:** 1Faculty of Health, Medicine & Life Sciences, University Maastricht, P.O. Box 616, 6200 MD Maastricht, The Netherlands; merelital@hotmail.com; 2Department of Pediatrics, Amsterdam UMC, University of Amsterdam, Vrije Universiteit, Emma Children’s Hospital, Meibergdreef 9, 1105 AZ Amsterdam, The Netherlands; E.F.G.Naninck@uva.nl; 3Department of Pediatrics—Division Neonatology, Erasmus MC University Hospital—Sophia Children’s Hospital, P.O. Box 2060, 3000 CB Rotterdam, The Netherlands; j.roelants@erasmusmc.nl (J.A.R.); i.reiss@erasmusmc.nl (I.K.M.R.); 4Department of Genetics, Erasmus MC University, P.O. Box 2060, 3000 CB Rotterdam, The Netherlands; g.vanderhorst@erasmusmc.nl (G.T.J.v.d.H.); i.chaves@erasmusmc.nl (I.C.); 5Department of Pediatrics—Amsterdam UMC—Emma Children’s Hospital, Meibergdreef 9, 1105 AZ Amsterdam, The Netherlands; h.vangoudoever@amsterdamumc.nl; 6Department of Pediatrics—Division Pediatric Intensive Care, Erasmus MC University Hospital—Sophia Children’s Hospital, P.O. Box 2060, 3000 CB Rotterdam, The Netherlands; k.joosten@erasmusmc.nl

**Keywords:** biorhythms, lactation, chrono-nutrition, circadian clock, diurnal variations, breast milk

## Abstract

Background: Breastfeeding is considered the most optimal mode of feeding for neonates and mothers. Human milk changes over the course of lactation in order to perfectly suit the infant’s nutritional and immunological needs. Its composition also varies throughout the day. Circadian fluctuations in some bioactive components are suggested to transfer chronobiological information from mother to child to assist the development of the biological clock. This review aims to give a complete overview of studies examining human milk components found to exhibit circadian variation in their concentration. Methods: We included studies assessing the concentration of a specific human milk component more than once in 24 h. Study characteristics, including gestational age, lactational stage, sampling strategy, analytical method, and outcome were extracted. Methodological quality was graded using a modified Newcastle-Ottawa Scale (NOS). Results: A total of 83 reports assessing the circadian variation in the concentration of 71 human milk components were included. Heterogeneity among studies was high. The methodological quality varied widely. Significant circadian variation is found in tryptophan, fats, triacylglycerol, cholesterol, iron, melatonin, cortisol, and cortisone. This may play a role in the child’s growth and development in terms of the biological clock.

## 1. Introduction

Human milk is the main source of nutrition for neonates. Breastfeeding has beneficial short- term and long-term health effects for both mother and child. It protects infants against mortality and morbidity from infectious diseases. It shortens recovery time during illness and is associated with improved cognitive development [1]. Accordingly, the World Health Organization recommends exclusive breastfeeding for the first six months of age and continued breastfeeding up to at least two years of age [2]. Human milk contains all (essential) nutrients required for growth and development, and it is responsible for the transfer of a plethora of bioactive factors such as hormones, immune factors, and possibly microbiota from mother to child [3]. Unlike formula feeding, the composition of human milk is variable in order to meet the infant’s physiologically-changing needs. Milk composition is different between mothers and populations, between term and preterm infants, and may even be different for boys and girls [4,5]. Additionally, some human milk components appear to change over the day. This suggests a diurnal or circadian rhythm, which is the focus of this study.

Like almost every living organism, humans exhibit an endogenous circadian clock that promotes survival by helping us anticipate predictable environmental changes such as light, temperature, noise, food, and exercise [6,7]. A selfsustained molecular oscillator, composed of clock genes, is located in virtually all cells of our body. This oscillator drives rhythmic expression of clock-controlled genes, which make up 10–20% of a tissues’ transcriptome [8]. This results in a nearly 24-hour rhythm, which results in circadian variations in hormone levels, enzyme activity, and cellular activity in most cells of the body [9]. To keep in phase with the 24-hour day-night cycle, the circadian system requires daily resetting by ‘zeitgebers’ of which light is the strongest. Photic information synchronises the master clock in the suprachiasmatic nucleus (SCN) in the hypothalamus. This is a process called ‘entrainment.’ The SCN, in turn, serves as a synchroniser for all peripheral oscillators [10]. Importantly, food intake can influence the phase of these peripheral oscillators [11]. Deregulation of the circadian clock can lead to health problems, such as sleeping disorders and metabolic diseases [7].

In the early postnatal period, the circadian clock is not yet fully functional [12,13,14,15]. However, the first signs of a circadian rhythm in the fetus can be observed from 30 weeks of gestation [16]. The synchronisation of the neonate’s circadian rhythm to its new environment outside the womb depends on external cues such as light/dark exposure and timing of feeding. Circadian fluctuations in human milk composition are likely to assist the transfer of information on time of day from the mother to her newborn [17]. This makes human milk a unique form of ‘chrono-nutrition’ [17] and possibly helps the neonate to synchronise with its external environment. If so, for babies fed with expressed (donor) milk, a mismatch between the time of expression and the time of feeding may have consequences for the development of their circadian clock and sleep homeostasis.

Various studies indicate circadian rhythmicity in human milk components, such as melatonin and cortisol [18,19]. However, a full overview of all human milk components that exhibit a circadian rhythm in concentration is currently lacking. A better understanding of the chronobiology of human milk could lead to essential recommendations such as timing of infant feeding and timing of milk expression. Furthermore, in human milk research, it could be relevant to introduce circadian variation as a potential confounder by correcting for timing of milk expression. This systematic review aims to give a full overview of the human milk compounds exhibiting circadian variation.

## 2. Materials and Methods

### 2.1. Literature Search

In advance, a detailed research protocol was written in adherence to the Preferred reporting items for systematic review and meta-analysis (PRISMA) guidelines, and submitted to the PROSPERO prospective register of systematic reviews (CDR42020187210) [20]. A comprehensive search strategy was developed in collaboration with a clinical librarian to search Embase, Medline (Ovid), Web of Science, Cochrane CENTRAL, and Google Scholar. Search terms were optimized for each specific database, as shown in Appendix A. The search was conducted on 25 May 2020. All articles resulting from the search were assessed for eligibility based on the titles and abstracts. Of the remaining articles, full text was screened to check for eligibility. If two articles used the same data, the article best describing the research process was included. Study selection, data extraction, and quality assessment were performed by two independent reviewers. Whenever necessary, a third independent reviewer was required to reach consensus.

### 2.2. Study Selection and Data Extraction

Only observational studies were included in this review. Studies were included if a specific human milk component was measured at two or more time points in a 24-hour period. Furthermore, a quantitative description of the outcome measure(s) was required for inclusion with concentrations of assessed human milk components being represented in relation to the time of day in order to give insight in circadian rhythmicity. For each milk component, we defined circadian variation as a significant difference in concentration between at least two time points within a 24-h period. Studies reporting on levels of xenobiotic or exogenous compounds in human milk were excluded. Animal studies were also excluded. If no full text was available and our personal request for full text did not result in any response from the author, the publication was excluded. The reference lists of all identified studies, reviews, and textbooks were reviewed in order to avoid missing relevant publications. No language restrictions or restrictions due to publication date were applied.

We created worksheets to systematically manage study selection, data extraction, and methodological quality assessment.

### 2.3. Methodological Quality Assessment

The Newcastle-Ottawa Scale (NOS) for quality assessment adapted for cross-sectional studies was used to grade the risk of bias (RoB) for each study [21]. According to the guidelines, the grading scale was adapted to our research question (See Appendix A). A population size of ≥10 mothers being studied for one day or ≥5 mothers being studied for two consecutive days was considered satisfactory and graded with one star. Selection criterium 4, known as the ascertainment of exposure, was adapted since all mothers were exposed to a circadian rhythm. It was replaced by an item scoring the number of time points or intervals per 24 h used for analysing the circadian rhythm. Outcome criterium 1, known as the assessment of the outcome, was modified to allow for discriminating the various analytical methods used to measure the human milk component of interest. With this method, studies were graded with a maximum of 10 stars, and study quality was considered adequate when 6 stars or more were reached.

## 3. Results

### 3.1. Study Selection

Upon removal of duplicates, a total of 2270 articles were identified by the search. Assessment of these articles based on titles and abstracts yielded a total of 146 publications of interest. Upon reading the full text of these publications, 64 additional papers were excluded. Reasons for exclusion in the selection process can be found in Figure 1. Reviewing the reference lists yielded one extra study meeting inclusion criteria, which resulted in a total of 83 studies assessing the circadian variation in human milk components to be included in our systematic review.

### 3.2. Study Characteristics

The included studies assessed different human milk components that could be classified in the following categories: macronutrients (carbohydrates, proteins, fats), micronutrients (vitamins, trace elements, and electrolytes), and bioactive factors (hormones, immune factors, and other bioactive factors). Studies were performed in different countries with the United States, Brazil, Australia, and Europe being most widely represented. The study population ranged from 1 to 200 mothers. Maternal age varied between 13 to 53 years even though maternal age was not reported by all studies. The majority of study populations included mothers who delivered at term either via cesarean section or vaginal delivery. Lactational stage varied widely with studies including colostrum samples from the first days postpartum to transitional or mature samples until the time of breastfeeding cessation. The timing of sampling varied widely between studies and included sampling at on-demand breastfeeding or sampling at set intervals, which range from 2 to 12 times in 24 h. The number of collection days ranged from 1 to 14 days. Analysed milk included foremilk (or prefeed), mid-milk, hindmilk (or post-feed), and pooled milk from one feeding. Heterogeneity among the studies was large, and sufficient suitable data was not available for any of the breast milk to perform meta-analyses.

### 3.3. Methodological Quality Assessment

The Risk of Bias score (RoB score) assigned to each study for methodological quality varied from two to ten stars with a median of six stars. An adequate grading of six stars or more was assigned to 45 of the studies (54) (See Appendix A). In 54 studies (65%), sample size was considered sufficient and, in 52 studies (63%), loss to follow-up was limited and described clearly. Regarding sampling frequency, two stars were assigned to 28 studies (34%) for sampling ≥ six times per day. One star was assigned to 30 studies (36%) for sampling three to five times per day and 25 studies (30%) did not receive any star for sampling only two times per day. Of the methods used for chemical analysis, 16 studies (19%) were graded two stars for using highly sensitive analytical methods such as mass spectrometry, 51 studies (61%) received one star, and 16 studies (19%) received no star for using an inaccurate analytical method or lacking a proper description. Highly sensitive analytical methods, which allow for detection of small changes in concentration, were predominantly used in the more recent papers. The majority of studies (*n* = 60, 72%) applied and described high standard statistics and 42 studies (51%) performed subgroup analyses or adjusted for at least one potential confounder.

### 3.4. Circadian Variation in Human Milk Components

We provide a full overview on circadian fluctuations in human milk components, grouped by physiological function, below. A full overview of the included literature on macronutrients, micronutrients, and bioactive factors including study characteristics, methodological quality assessment, and outcome can be found in Appendix A. The main results are summarised in Table 1. Based on the high heterogeneity among studies, quantitative meta-analysis was not considered feasible for any of the components. If multiple studies with high methodological quality reported consistent patterns, the estimated timing of the peak were visualized in a semi-quantitative schematic “info-graph” (See Figure 2).

#### 3.4.1. Macronutrients

##### Carbohydrates

Carbohydrates serve as an important energy source for neonatal metabolism, especially for the brain [22]. As shown in three independent studies of sufficient quality [23,24,25], total carbohydrate concentration in human milk does not show any circadian variation. For glucose, more research is necessary since the only two studies [26,27] on this component are contradictive.

Lactose content of human milk did not show a circadian rhythm in most studies [26,27,28,29,30]. The only included study reporting circadian variation, which had the least Risk of Bias, did not find a consistent pattern [28].

The carbohydrates glucose 6-phosphate, glucose 1-phosphate, UDP-glucose, and UDP-galactose [26] do not show circadian fluctuation, but evidence is limited with only one included study on each component. Overall, carbohydrates do not seem to show circadian variation in human milk.

##### Proteins

Proteins contain (essential) amino acids, which serve as building blocks for many neonatal anabolic pathways. Furthermore, they are an important source of energy [31]. The total protein content of human milk does not exhibit circadian variation in nine out of eleven studies [23,24,25,28,29,32,33,34,35]. In the two studies [27,36] that do indicate a circadian rhythm in total protein concentration, the acrophase (peak) is discordant and, therefore, it potentially indicates false-positive findings. Studies investigating total nitrogen, which is a proxy for protein, found no consistent circadian pattern [36,37,38].

Several studies focused on specific proteins and enzymes. The amide urea did not show circadian variation in one study [37]. Only one study with inconclusive results could be included on the neuropeptide delta-sleep-inducing peptide (DSIP) [39]. In addition, the enzymes’ serum-stimulated lipase (SSL), lipoprotein lipase (LPL), superoxide dismutase, glutathion peroxidase 3, and amylase [38,40,41,42] were only examined in one study and the enzyme bile salt-stimulated lipase (BSSL) was only examined in two studies [38,40]. Although these studies did not show circadian variation, evidence is too limited to draw conclusions on the existence of circadian variation in these human milk proteins.

Three studies assessed the presence of circadian variation in individual amino acids [18,37,43]. Of these, one study with adequate methodology [43] suggests circadian variation in the concentration of tryptohan, methionine, histidine, phenylalanine, and tyrosine. However, only the diurnal rhythm of tryptophan with its peak in the early morning is confirmed in a second study of sufficient quality [18] (See Figure 2). The circadian variation of the other amino acids seems to be more pronounced in mature milk than in colostrum and transitional milk, but was only reported in one [36] of three studies [36,37,38]. Overall, although total protein concentrations in human milk do not seem to exhibit a circadian pattern, some individual amino acids may show circadian variation.

##### Fats

The fat content of human milk accounts for ±50% of its total energy supply and is the supplier of essential fatty acids required for the central nervous system and retina development [22]. Out of the 19 included studies on total fat content variation in human milk [23,27,44,45], 15 report circadian variation in total fat concentration [24,25,29,32,38,46,47,48,49,50,51,52,53,54,55]. In the majority of studies, the acrophase was found in the evening [25,32,47,48,50,51,52,53,55] (See Figure 2). Regarding the statistical methods in the affirming studies, 10 were performed and described properly.

One study found that circadian variation in the concentration of total fat decreased with increasing lactational stage [47], whereas two other studies did not find an effect of lactational stage on circadian variation [44,51]. None of the studies on circadian variation in fat concentration performed subgroup analyses for foremilk versus hindmilk samples despite this being known to affect fat concentration in human milk [29].

Out of the studies not reporting a circadian rhythm in total fat, only one study compared daytime milk versus nighttime milk, which could lead to missing an existing circadian variation [44]. In another study, the sample size was rather small with only four mothers delivering milk samples suited for analysis [24]. Lastly, one study did not find circadian variation in total fat content, but did find a non-significant tendency of fat concentration to be higher at 10:00 h [27].

Overall, despite the fact that various factors such as maternal diet, duration of lactation, and pre- versus post-feed affect the fat content of human milk [28,29,47,56], most studies indicate the presence of circadian variation in total fat concentration of human milk with the acrophase in the evening.

For triacylglycerols, which make up more than 95% of total lipids [57], findings are clearly in line with total lipids, as two [56,58] of three studies [40,56,58] report significant circadian variation with its nadir (lowest value) in the morning and its acrophase either in the afternoon or evening. The same can be concluded for cholesterol, whereby three [48,56,59] out of four studies [27,48,56,59] found circadian variation with the acrophase tending to be in the evening. In contrast, in six studies, the concentration of individual free fatty acids consistently did not show a circadian rhythm in six studies [27,32,56,60,61,62]. This might be explained by the small sample sizes. Since only one study investigated sphingomyelin phospholipids and fat globules, it is not possible to draw a conclusion on rhythmicity in these human milk lipids [56,63,64].

#### 3.4.2. Micronutrients

##### Vitamins

Overall, vitamins A, B1 (thiamine), B2 (riboflavin), B3 (niacin), B6 (pyridoxine), B8 (biotin), B12 (cobalamin), vitamin E (tocopherol), and choline do not demonstrate circadian variation [27,33,63,65,66,67,68], whereas vitamin B11 (folate) shows circadian variation in one study [69]. However, the evidence is limited and, therefore, results on the presence of circadian variation in human milk vitamins are inconclusive.

##### Trace Elements and Electrolytes

The trace-element iron consistently shows circadian variation in concentration with the acrophase during the evening or night [53,67,70,71] (See Figure 2). One study found that circadian variation in iron concentration was suppressed in mothers with iron deficiency [67]. Sodium and potassium levels possibly exhibit a circadian rhythm, based on a single publication with adequate methodology [72] and an additional exploratory study in which a false positive result cannot be ruled out since no correction for multiple testing was made [35]. Calcium concentrations in human milk do not exhibit circadian variation [35,73,74,75], and the same result was found for copper [53,70,71,75]. Zinc concentrations in human milk might, however, follow a circadian rhythm with three [53,74,76] out of seven [53,70,71,74,75,76,77] publications using the same adequate analytical method finding its acrophase in the morning. For phosphorus, magnesium, CuZn-superoxide dismutase, iodide, iodine, and molybdenum, evidence is either contradictive or limited [35,44,73,74,75,78,79,80].

Overall, the presence of circadian variation in iron concentration is convincing, while no strong evidence exists for other trace-elements in human milk.

#### 3.4.3. Bioactive Factors

##### Hormones

From all endocrine factors in human milk, melatonin is by far the most studied in relation to rhythmicity. Melatonin is an important indicator of the night, but also functions as an antioxidant, anti-inflammatory agent, an antinociceptive agent, and immune regulator [81]. It has a profound circadian pattern in adult serum. In human milk, melatonin also exhibits circadian variation [41,82,83,84,85,86,87,88,89] with the acrophase consistently at night with a mean peak value of 46.9 ± 4.2 pg/mL (mean ± SEM) in four studies measuring at midnight, and undetectably low levels during the day [83,86,87,89] (See Figure 2).

Glucocorticoids, which are adrenal steroids important for regulation of the stress-response, exhibit a circadian rhythm in adults, which can already be detected in children from one month of age onwards and continues to develop during the first year of life [14]. In human milk, the glucocorticoid cortisol and its inactive form cortisone also show circadian variation with the highest concentration consistently found in the morning [24,89,90,91,92,93,94] (See Figure 2).

For leptin, prolactin, and the parathyroid hormone-related protein (PTHrP), evidence is too limited, but some studies did show circadian variation in these endocrine human milk components as well [28,95,96]. In conclusion, the human milk hormone concentrations currently studied show circadian variation with a specific pattern identified for each hormone.

##### Immune Factors

Many human milk health benefits, such as protection from infectious diseases, are related to human milk containing various immune factors. For the immunoglobulins IgA, IgG, and IgM, the presence of circadian variation is inconclusive, as findings are contradictive in the reported timings of the highest and the lowest concentration levels [35,44,97,98]. Similarly, no conclusions can be drawn on the existence of circadian variation in cytokine levels. Two studies on interleukins show contradictory results, which might be partly caused by the application of different analytical methods [86,99]. For tumor necrosis factor (TNF)-α [86,87,99] and transforming growth factor (TGF)-β [35,89], results are also inconclusive. Most publications on interferon (IFN)-γ demonstrate the existence of circadian variation, but the course of this variation is contradictory. Results on lactoferrin are also inconclusive [97,100]. Only separate studies were included on complement factors C3 and C4, lysozyme, and the epidermal growth factor [44,98,101].

Interleukin (IL)-2 levels in colostrum of mothers who gave birth by vaginal delivery were higher at 00:00 h compared to 12:00 h. However, this was not apparent in the colostrum of mothers when giving birth by cesarean section [86]. Mode of delivery, however, did not affect the circadian variation in the colostrum levels of other interleukins (IL-10, IL-4, or IL-5). 

Overall, no strong evidence of an existing circadian variation in human milk immune factors is found in the included literature.

##### Other Bioactive Factors

Next to various macronutrients, micronutrients, hormones, and immune factors, human milk contains several other bioactive factors. For some of these, the circadian pattern in their concentrations has been assessed. Nucleotide concentrations in human milk could exhibit circadian variation based on one out of two relevant studies [102,103]. Two studies on rhythmicity of antioxidant activity showed contradictive results [41,104]. Similarly, no conclusions can be drawn on the presence of a circadian rhythm in carotenoids, micro RNAs, human milk oligosaccharides, citrate, malondialdehyde, perchlorate, and thiocyanate in human milk [26,30,67,78,105,106].

## 4. Discussion

This systematic review provides an overview of the current literature on circadian variation in human milk components. Taking into account the characteristics and methodological quality of the studies, there is strong evidence of circadian variation in human milk tryptophan, fats, triacylglycerol, cholesterol, iron, melatonin, cortisol, and cortisone. There is also convincing evidence of no circadian variation in the carbohydrate and total protein content of human milk. For other human milk components, the current evidence is too limited to draw conclusions. For many components, the evidence was inconclusive or even contradictory due to limitations in experimental design.

Most study populations were small and heterogeneous regarding maternal characteristics. Since milk composition varies considerably between mothers based on dietary intake and parity, there may have been a dilution of effects as well as false negative results. In the majority of studies, potential confounders were not taken into account. These concern (i) maternal characteristics and (ii) the sampling strategy. Maternal characteristics that can impact circadian rhythmicity in human milk composition include age, infection [87], hypertension [35], nutritional deficiencies such as iron deficiency [67], and duration of lactation [35,36,39,43,44,89,98,99,107]. A factor that needs further exploration in this context is a mode of delivery. The findings of circadian variation in IL-2 levels in colostrum after vaginal delivery, but not after cesarean section, suggests that the mode of delivery can be a relevant factor [86]. This is the most relevant for studies performed in countries with high cesarean section rates, such as Brazil [86,108], whereby lack of rhythmicity may not be generalized to the general population worldwide. A second important issue concerns the sampling strategy. Frequency and timing of milk sampling varied widely between studies. With a limited amount of time points per day as seen in nearly a third of the studies, circadian variation can easily be missed. The acrophase and nadir are difficult to establish. Another limitation is the short window of observation, which was limited to one day per mother in the majority of the studies. Subgroup analyses in colostrum, transitional, and mature milk was done in only a few studies. Changes in circadian variation in mature milk over longer time periods was not explored.

### 4.1. Strengths and Limitations

To the best of our knowledge, this is the first comprehensive systematic review of studies on circadian rhythmicity in human milk components. A strength of our analysis is the broad search strategy and systematic quality assessment in accordance with current guidelines. However, this study does have some limitations. For feasibility, we excluded all animal studies. However, primate studies may reveal evidence that can be extrapolated to human milk. Our attempt to summarise results for each specific component were complicated by high heterogeneity among studies regarding populations and methodology. For example, our estimation of the timing of the acrophase can be biased due to differences in daylight and darkness timing in different countries and seasonal fluctuations thereof.

### 4.2. Physiological Role of Circadian Rhythmicity in Human Milk

A question that remains to be addressed is whether circadian variation in human milk composition is defined by the mothers’ intrinsic circadian clock in mammary gland tissues, or by external or behavioral factors, such as sleep and timing of food intake, or a combination thereof. It is also unclear to what extent ethnicity, socioeconomic status, and life style factors are relevant. The mammary epithelial cells are capable of active receptor-mediated transport of molecules such as iron [109]. This explains why fluctuations in milk components do not always reflect fluctuations in maternal blood concentrations. A study comparing plasma to milk variations in healthy nursing women found that the acrophase of malonic dialdehyde (MDA) and α-Tocopherol were synchronous while the acrophase of iron was found 6 h later in milk (12:00) than in blood (18:00 h) [67].

The evolutionary development of circadian rhythmicity in human milk is likely to have a beneficial nutritional and metabolic effect on the developing infant. This is in line with changes in human milk within one feeding and over the courses of lactation to meet the infant’s changing nutritional needs. The high levels of fat with its main constituent triacylglycerol during the morning as compared to the evening are likely to synchronize with the child’s fat metabolism and, hence, act as an important cue for healthy growth. The circadian variation in fat may even play a role in lowering the risk for obesity and other cardiovascular risk factors, known to be associated with breastfeeding as compared to formula feeding [6]. In accordance, the absence of circadian variation in carbohydrates and total protein could also be beneficial to the newborn, as it may reflect the continuous high demands for these components during early postnatal life, which is required for healthy brain function and development. Another physiological role of circadian rhythmicity in human milk may be assisted by the development and entrainment of the child’s circadian rhythm. The fact that the acrophase of tryptophan, a metabolic precursor of melatonin [110], precedes the acrophase of melatonin by several hours, suggests the presence of a balanced timing system. A role for human milk in the development of this rhythm was suggested by a recent study showing improved circadian rest-activity rhythm in breastfed infants compared to mixed-fed infants [111]. These findings also suggest that a mismatch between the time of milk expression and time of feeding possibly disrupts these processes and affects infant growth and the development of their sleep-wake cycle. It is unclear whether this can be relevant for infants who receive their own mother’s milk or donor milk. Further research is required to discover whether disturbance of maternal circadian rhythm, such as by artificial lighting and the 24/7 economy, or even stronger, by maternal jet lags or night shifts, impacts the circadian rhythmicity of milk composition and the child’s circadian rhythm. Even if this may play a role, we expect that breast feeding will remain superior to formula feeding, which shows no variation at all, and lacks many of the unique nutritional, hormonal, and immunological characteristics of human milk.

## 5. Conclusions

Circadian variation in human milk composition is found in various components. To deepen our understanding of the chronobiology of human milk, high quality studies are required to examine the many understudied human milk components such as insulin, prebiotics, probiotics, immune cells, and metabolites. Factors that determine circadian rhythmicity, such as the clock genes and environmental factors, should be addressed. Furthermore, sampling should be performed frequently, ideally ≥ four time points per 24 h and equally spread over the day. For human milk researchers, our data give direction for deciding whether or not to correct for milk expression timing in future analyses. Future research should also focus on the importance of human milk circadian rhythmicity and disturbance thereof for child health, behaviour, and development. In conclusion, despite the limitations, the accumulated evidence indicates that at least several essential human milk components systematically vary over the day by identifying human milk as chrono-nutrition in early life.

## Figures and Tables

**Figure 1 nutrients-12-02328-f001:**
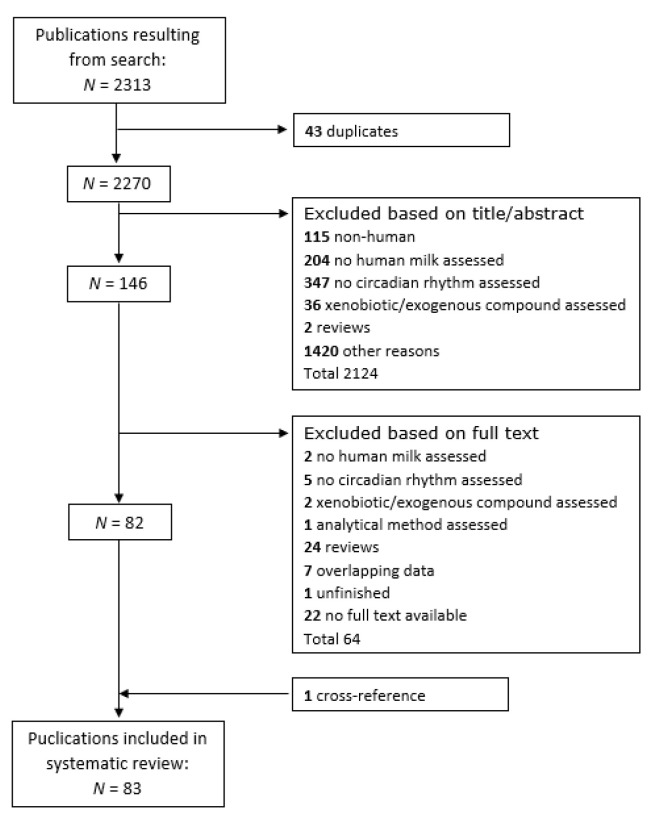
Flowchart of a study selection based on the inclusion and exclusion criteria.

**Figure 2 nutrients-12-02328-f002:**
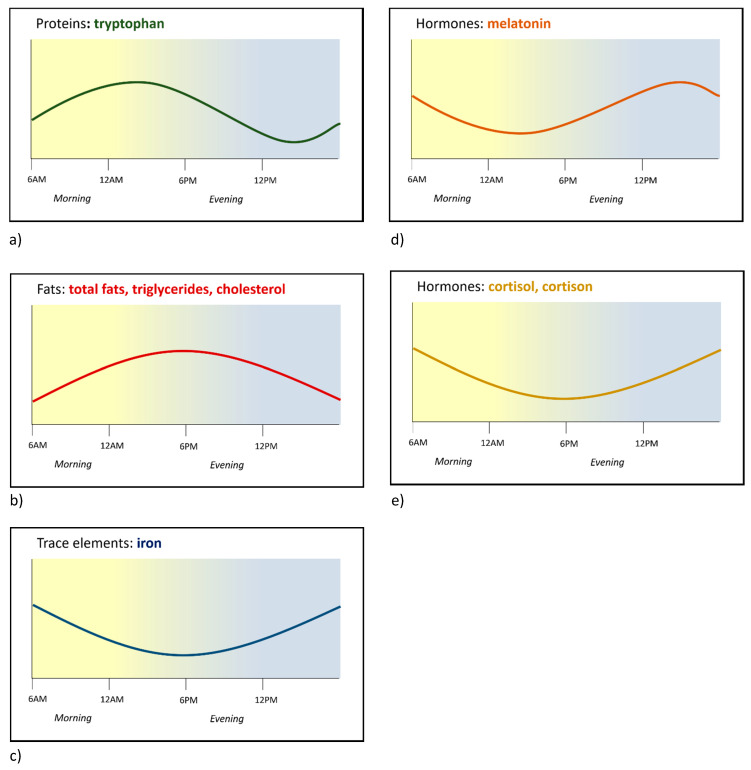
Circadian variation in human milk components. Schematic presentation of estimated circadian curves based on peak and trough values reported in the literature. Circadian variation curves are shown for the following components: (**a**) tryptophan [18,43], (**b**) total fats, triglycerides, cholesterol [25,32,47,48,50,51,52,53,55,56,58,59], (**c**) iron [53,67,70,71], (**d**) melatonin [41,82,83,84,85,86,87,88,89], and (**e**) glucocorticoids [24,89,90,91,92,93,94].

**Table 1 nutrients-12-02328-t001:** Summary of evidence on circadian variation in human milk components, showing conclusions per compound group.

Compound Group	Circadian Variation Identified	No Circadian Variation Identified	Inconclusive
Carbohydrates		Carbohydrates	Lactose, glucose, glucose 6-phosphatase, glucose 1-phosphate, UDP-glucose, UDP-galactose
Proteins	Individual amino acids i.e., tryptophan	Total protein, BSSL, total nitrogen	Urea, cobalamin, delta-sleep-inducing peptide, serum-stimulated lipase, lipoprotein lipase, superoxide dismutase, glutathion peroxidase 3, amylase
Fats	Fats, triacylglycerol, cholesterol	Fatty acids	Sphingomyelin and phospholipids
Vitamins		A, B1-3, B6, B8, B11, B12, E, choline	
Trace elements and elektrolytes	iron	Ca, Cu	Zn, P, Mg, CuZn-SOD, I2, I-, molybdenum
Hormones	Melatonin, cortisol, cortison		Leptin, prolactin, PTHrP
Immune factors			IgA, IgG, IgM, cytokines, interleukins, TNF-α, TGF-β, IFN-γ, lactoferrin, C3, C4, lysozyme, EGF
Other bioactive factors			Nucleotides, antioxidant activity, carotenoids, miRNA’s, citrate, malondialdehyde, perchlorate, thiocyanate, oligosaccharides

UDP: Uridine diphosphate. BSSL: Bile salt-stimulated lipase. PTHrP: parathyroid hormone related protein. Ig: immunoglobulin. TNF-α: tumor necrosis factor-α. TGF-β: transforming growth factor. IFN-γ: interferon- γ. HMO: human milk oligosaccharides. SOD: superoxide dismutase.

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
