# Peer review of "Circadian Variation in Human Milk Composition, a Systematic Review"

_nutrients, 2020, doi:10.3390/nu12082328_

Round 1

Reviewer 1 Report

Review: Circadian variation in human milk composition, a systematic review 

Introduction:

Concisely summarized latest findings of field and outlined need for current study.

Methods:

Line 88 – Authors could briefly outline the search criteria/code, or include it as supplementary material.

Authors could add a paragraph outlining their circadian variation measurements – what is the variation in a nutrient concentration that is considered “circadian variation” between time points? Was this comparison performed for each value/concentration? How was it calculated for each nutrient?

Results:

Line 125 – Authors could include more details on the “search”, e.g. terminology used across platforms.

Line 200-201 – Authors write that one study shows circadian variation in select amino acids, but reference three studies. It would be informative to indicate the specific study referenced in the text as it is mentioned.

Line 203 – “second study” not referenced. More clarity and attention to such writing standards and etiquette are required.

Line 205 – “reported in one study”, but two are referenced.

Line 304 – Please include the evidence for variation in the table – e.g. the average concentration values in day/night milk samples. The presentation of data would be strengthened by inclusion of the number of studies reported on for each nutrient, and the total sample number compared overall for each nutrient.

The Table could be improved upon with additional details.

ine 311 – What is the y-axis on the graph? Similar to above, inclusion of number of samples/studies would be beneficial. Are the values plotted on curves normalized/averaged? How is the curve plotted?

Discussion:

Some, not all, study limitations are discussed. Assessment and interpretation is to be welcomed here.

Could the differences seen in milk composition be due to infants’ nutritional needs? i.e. milk expressed for preterm infants. [Lubetzky, R., Littner, Y., Mimouni, F.B., Dollberg, S. and Mandel, D., 2006. Circadian variations in fat content of expressed breast milk from mothers of preterm infants. Journal of the American College of Nutrition25(2), pp.151-154.]

Self-citation: Author Reiss cites himself once, in Ref #6. Normal.

Overall:

Minor typos throughout. More grammatical hygiene is required by the overall presentation. E.g.

Line 243, spelling of “elektrolytes” ?

Line 256 …for other trace-element in human milk. INSERT ‘s’

Line 276 are related to the fact that human milk containsing various immune factors

Line 283 interferon (IFN)-ꙋ  What is the strange iconography following the hyphen?

Line 289 Mode of delivery, however, did not affect  INSERT commas

Line 296 Nucleotides concentrations in human milk…  Nucleotide is an adjective , remove ‘s’

Line 353 what extend ethnicity, Clear evidence that not one single co-author nor the senior author has taken the effort to proof-read their manuscript with MS Word using its powerful spell check functions of green and red alert warnings: EXTENT

END or REVIEW.

Reviewer 2 Report

This comprehensive review does provide an interesting perspective in our understanding of breast milk composition, although the conclusions of the review are somewhat underwhelming.  This is likely the result of heterogeneity between studies included in this review. Overall, the value of this review is that it lays out a foundation to potentially build more comprehensive studies to test circadian effects on nutrients in human milk.  

Specific points:

-Line 52: Although there is research suggesting the there is a milk microbiota, this is disputed amongst the breast milk research field, as no comprehensive examination has identified bacterial cells in breast milk prior to suckling.  Suggest changing this sentence to be more suggestive.

-Line 145: 'The number of sampling days...1-10'.  Suggest providing a sentence or two about this being a limitation in circadian fluctuation studies since it is a very small window of analysis given the totality of breastfeeding duration.  Additionally, does this suggest that sampling days were between 1-10 days at different points of breastfeeding duration?  If so, this should be addressed in the limitations as well, given the drastic changes in volume over breastfeeding duration.

-Line 283: IFN does not have a proper designation.  Please change.

-Line 287: Authors indicate the literature showed IL-2 differences between vaginal and cesarean sections.  There is also literature showing differences in vaginal versus cesarean for TNFalpha and other proinflammatory cytokines (PMC6214518).

-Line 331-332: Same as above.

Reviewer 3 Report

This is an interesting paper which highlights the likelihood of changes in some breastmilk components across a 24 hour day.

The systematic review appears to be methodologically appropriate.

I am concerned about the final statement in the abstract:

This may have consequences for infants who receive expressed own mother’s milk or donor milk when the timing of expression is not matching with the timing of feeding.

Providing expressed breastmilk is a difficult process for individual mothers and expecting mothers to ensure that expressed breastmilk fed to their baby matches the time of day it was expressed, will put even more pressure on mothers.

Expressed breastmilk supplied to milk banks is always pooled for pasteurisation so it would be very difficult, not it would be impossible, to ensure that breastmilk supplied would match the time of the day the baby was being fed.

I think the authors should think carefully about what they say about the timing of the use of expressed breastmilk. They may be creating an expectation that can never be met.

Small edit

The following sentence in the abstract doesn't make sense - the English needs changing:

Human milk changes over the courses of lactation, feeding as well as the day in order to perfectly suit the infant’s nutritional and immunological needs.
